# Designing a comprehensive behaviour change intervention to promote and monitor exclusive use of liquefied petroleum gas stoves for the Household Air Pollution Intervention Network (HAPIN) trial

Kendra N. Williams [1,2] Lisa M. Thompson,[3] Zoe Sakas,[4] Mayari Hengstermann,[5] Ashlinn Quinn,[6] Anaité Díaz-Artiga,[5] Gurusamy Thangavel,[7] Elisa Puzzolo,[8] Ghislaine Rosa,[4] Kalpana Balakrishnan,[7] Jennifer Peel,[9] William Checkley,[1,2] Thomas F Clasen,[10] J Jaime Miranda [11] Joshua P Rosenthal,[6] Steven A Harvey,[12] on behalf of the Household Air Pollution Intervention Network (HAPIN) trial Investigators

KNW and LMT are joint first authors.

For numbered affiliations see end of article.

**Correspondence to**
Dr Kendra N. Williams;
kendra.williams@jhu.edu

## ABSTRACT

**Introduction** Increasing use of cleaner fuels, such as liquefied petroleum gas (LPG), and abandonment of solid fuels is key to reducing household air pollution and realising potential health improvements in low-income countries. However, achieving exclusive LPG use in households unaccustomed to this type of fuel, used in combination with a new stove technology, requires substantial behaviour change. We conducted theory-grounded formative research to identify contextual factors influencing cooking fuel choice to guide the development of behavioural strategies for the Household Air Pollution Intervention Network (HAPIN) trial. The HAPIN trial will assess the impact of exclusive LPG use on air pollution exposure and health of pregnant women, older adult women, and infants under 1 year of age in Guatemala, India, Peru, and Rwanda.

**Methods** Using the Capability, Opportunity, Motivation–Behaviour (COM–B) framework and Behaviour Change Wheel (BCW) to guide formative research, we conducted in-depth interviews, focus group discussions, observations, key informant interviews and pilot studies to identify key influencers of cooking behaviours in the four countries. We used these findings to develop behavioural strategies likely to achieve exclusive LPG use in the HAPIN trial.

**Results** We identified nine potential influencers of exclusive LPG use, including perceived disadvantages of solid fuels, family preferences, cookware, traditional foods, non-food-related cooking, heating needs, LPG awareness, safety and cost and availability of fuel. Mapping formative findings onto the theoretical frameworks, behavioural strategies for achieving exclusive LPG use in each research site included free fuel deliveries, locally acceptable stoves and equipment, hands-on training and printed materials and videos emphasising relevant messages. In the HAPIN trial, we will monitor and reinforce exclusive LPG use through temperature data loggers, LPG fuel delivery tracking, in-home observations and behavioural reinforcement visits.

**Conclusion** Our formative research and behavioural strategies can inform the development, implementation, monitoring and evaluation of theory-informed strategies to promote exclusive LPG use in future stove programmes and research studies.

**Trial registration number** NCT02944682, Pre-results.

## Strengths and limitations of this study

► Application of the Capability, Opportunity, Motivation–Behaviour framework and Behaviour Change Wheel facilitated the identification of context-specific influencers of fuel choice.

► The theory-guided formative research methods enabled the development of tailored behavioural strategies to promote exclusive use of liquefied petroleum gas (LPG).

► Our formative research did not consider market forces or costs of LPG given the intent to inform a trial in which LPG will be delivered for free.

► The extensive behavioural monitoring and reinforcement protocol may not be feasible for replication in all contexts.

## INTRODUCTION

Nearly three billion people worldwide use solid fuel (wood, charcoal, dung, crop residue or coal) and kerosene for cooking, heating and lighting.[1] Use of these fuels leads to high levels of household air pollution (HAP), resulting in negative impacts on

health, environment, well-being and climate.[2] Substitution of cleaner-burning fuels such as liquefied petroleum gas (LPG) has the potential to mitigate these negative outcomes.[3]

Stove programmes and research studies have focused on improved cookstoves (eg, rocket or vented chimney stoves[4–8] and cleaner fuels (eg, pellet,[9] ethanol[10 11] and LPG[12 13] to reduce exposure to fine particulate matter ($PM_{2.5}$) and carbon monoxide (CO) and subsequently improve health.[14] However, most programmes report limited exposure reductions (postintervention 24–48 hour mean $PM_{2.5}$ kitchen concentrations range from 120 to $280\,\mu g/m^3$ for cleaner fuel stoves and $290–410\,\mu g/m^3$ for improved solid fuel stoves) and uncertain health benefits.[15] One of the main reasons for this is the continued use of solid fuel stoves alongside cleaner fuel stoves—a practice known as stove stacking.[14] Models indicate that just 1 hour of traditional stove use per week can raise exposure to $PM_{2.5}$ and CO above the WHO recommended interim target of $35\,\mu g/m^3$ for annual mean $PM_{2.5}$ concentration.[16 17] To reach this target, many programmes are shifting away from improved solid fuel stoves towards promoting exclusive use of cleaner fuels. Equally important is the abandonment of solid fuel stoves.[14 18 19]

Cleaner fuel options for low-income and middle-income countries (LMICs) include LPG, electricity, piped natural gas, alcohol and biogas. However, electric stoves are not yet a viable option in regions with small, unreliable electric grids, piped natural gas is not widely available, alcohol fuel supply is typically limited and biogas is a high-maintenance option for rural settings.[20 21] LPG is a viable, scalable cleaner fuel option, however there are significant barriers to sustained, exclusive LPG use in LMICs.[20] The primary barrier is cost: poor families often cannot afford to purchase LPG stove or refill gas cylinders.[20 22–24] Another major barrier is access, especially in rural areas where the LPG supply infrastructure is limited.[20 24] Markets that assure adequate supply to meet household demand are a critical need.[25–28] At the household level, other factors play a role, including perceptions that traditional foods prepared with a solid fuel stove taste better,[29–31] and that two-burner LPG stoves cannot accommodate cooking large quantities of food.[32] Finally, fear that LPG stoves are dangerous may impede adoption.[20]

While overcoming behavioural barriers is critical to achieving long-term use of cleaner cookstoves and fuels, programmes and research studies that have integrated behavioural components into their campaigns often lack theoretically grounded and context-specific formative research on behavioural factors influencing exclusive use.[14 18 33] Analytical frameworks and conceptual models such as the Risks, Attitudes, Norms, Abilities, and Self-regulation,[34] Capability, Opportunity, Motivation–Behaviour (COM–B) and Behaviour Change Wheel (BCW) can guide the development and implementation of behavioural interventions,[35–37] but have not been widely used to promote sustained, exclusive use of cleaner cookstoves and fuels.[38–41]

We sought to overcome these barriers within the Household Air Pollution Intervention Network (HAPIN) trial.[42] The HAPIN trial aims to measure the effect of an LPG cooking intervention on HAP and health among study populations in Guatemala, India, Peru and Rwanda. Using a randomised controlled design, HAPIN will enrol 800 pregnant women (9 to <20 weeks gestational age) and up to 200 older adult women residing in the same homes in each country. Participants in the intervention group will receive an LPG stove and two LPG cylinders, approximately 18 months of free LPG deliveries, stove repairs as needed and continuous cooking behaviour-change support provided by local field staff. Primary outcomes are low birth weight, stunting and severe pneumonia in children less than 1 year of age, and blood pressure in older women.[42 43]

Achieving exclusive LPG use and abandonment of solid fuel stoves are essential to reduce HAP exposures within the HAPIN trial. In this paper, we describe formative research guided by the COM–B, BCW and Theoretical Domains Framework (TDF) to develop locally adapted behavioural strategies for promoting exclusive LPG use within the HAPIN trial.[36 37 44] We first present a comparison of key findings from formative research activities related to perceptions and use of LPG across the four research sites. We then discuss how we applied findings to develop behavioural interventions designed to achieve exclusive LPG use. We conclude with a protocol outlining the strategies we will use to monitor and reinforce LPG use in the main HAPIN trial.

## METHODS
### Guiding principles for behavioural strategies
The HAPIN research team formed a Behavioural and Economics Core (BEC) to address behavioural components of the trial. The BEC includes representatives from each participating country and health behaviour experts who provide guidance. The BEC concluded that behavioural strategies would require adaptation to contextual differences of each site, but strategies should share a common set of guiding principles, including:
1. Provide appropriate training on proper use and maintenance of LPG stoves and equipment to ensure safe operation.
2. Address context-specific barriers and facilitators to sustained, exclusive use of LPG and abandonment of traditional solid fuel stoves.
3. Maximise exclusive LPG use and minimise use of solid fuels among intervention households.
4. Monitor solid fuel stove use and reinforce exclusive LPG use in intervention households that continue to use solid fuels for cooking.
5. Avoid emphasising potential health benefits of LPG to minimise the risk of introducing bias when participants report health outcomes.

### Formative research
#### Theoretical grounding
We used the COM–B and BCW to guide the design of formative research activities and to apply findings to the

development of behavioural interventions. The COM–B model is a behavioural system that provides a foundation for evaluating the capabilities, opportunities and motivations that drive behaviour, highlighting that a 'behavioural diagnosis' must be understood to develop effective interventions.[37] The components of the COM–B map onto the theoretically derived determinants of behaviour from the TDF.[35] The TDF is comprised of 14 theoretical domain functions (Knowledge; Skills; Social/Professional Role and Identity; Beliefs about Capabilities; Optimism; Beliefs about Consequences; Reinforcement; Intentions; Goals; Memory, Attention and Decision Processes; Environmental Context and Resources; Social Influences; Emotions; and Behavioural Regulation) synthesised from 33 theoretical models and 128 constructs derived from these models.[35 45] The BCW includes nine intervention functions (education, persuasion, incentivisation, coercion, training, enablement, modelling, environmental restructuring and restrictions) that can be applied to address gaps in identified capabilities, opportunities and motivations to promote behaviour change.[36 37] Using the COM–B, TDF and BCW frameworks, we selected relevant domains and functions to develop behavioural strategies that are contextually specific (across HAPIN research sites) and grounded in theory.

### Study sites

Formative research was conducted in rural communities of Jalapa Department, Guatemala; Puno Province, Peru; Kayonza District, Eastern Province, Rwanda; and Nagapattinam and Kallakurichi (previously Villupuram) Districts in Tamil Nadu, India. Households in these rural communities were located between 30 min and up to several hours from main cities and varied in population density (Jalapa Department density $170/km^2$; Kayonza District density $180/km^2$; Puno Province density $18/km^2$; Nagapattinam District density $615/km^2$ and Kallakurichi District density $480/km^2$). Formative research surveys conducted in the communities found LPG stove ownership to be 0% in Rwanda, 68% in Peru,[46] 31% in Guatemala and 57% in India. However, exclusive use of LPG stoves was lower: 0% in Rwanda, 3.5% in Peru[46] and 7% in Guatemala. In India, only 29.5% primarily used LPG for cooking (data on exclusive LPG use were not available).

### In-depth interviews, rapid assessments and focus group discussions

In-depth interviews (IDIs) were conducted using semi-structured interview guides, tailored for each research site (table 1). Participants were selected based on the following criteria: lived in a rural community in the country-specific study site, female, between the ages of 18 and 68 and able to understand and provide consent. In each site, we aimed to include participants with and without previous knowledge and/or use of LPG. Teams in India and Guatemala also sought to include some men meeting the same criteria. The following themes were covered during IDIs:

1. Stoves owned and frequency of use.

2. Preferred stoves for traditional dishes and beverages.
3. Family influences on stove and fuel use.
4. Temporal, seasonal, and circumstantial influences on stove choice.
5. Perceived benefits and disadvantages of traditional stoves.
6. Knowledge and perceptions of LPG stoves.
7. Reasons for stove stacking.
8. Fuel purchase and solid fuel collection practices.
9. Perceived impact of LPG on daily life and household status.
10. Cooking tasks and consumption patterns, including during pregnancy and after birth.

In Rwanda, cooking demonstrations and food tasting tests were conducted prior to IDIs in participating homes who lacked exposure to LPG. Several focus group discussions (FGDs) were conducted in Rwanda to develop materials given minimal familiarity with LPG in the study area.

Behaviour change materials were developed based on IDI and FGD findings by local teams. FGDs were then conducted with participants in Rwanda, India and Peru according to the same eligibility criteria as IDIs to review draft materials (table 1). Participants were asked to describe their understanding of the messages being conveyed, any barriers and facilitators to LPG use not captured, whether messages could be understood based solely on the pictures (given low literacy rates) and whether participants felt represented by the images. Given the extensive governmental support and ubiquity of LPG in India, FGDs aimed to identify a minimum set of information necessary for promoting exclusive LPG use among intervention households to minimise contamination bias in control households. Materials were modified based on FGD feedback.

### Pilot studies with LPG cooking equipment

Following development of the behaviour change materials, we conducted pilot studies to test and revise procedures for the main trial, test the effectiveness and acceptability of the behavioural strategies and estimate anticipated $PM_{2.5}$ and black carbon levels (HAP results will be published separately).[47] Eligibility criteria included female, primary cook, 18–34 years of age, lived in a rural community in the country-specific study site, pregnant (<20 weeks gestation), non-smoker and reliance on biomass fuel for cooking. Women in India (n=40), Rwanda (n=40) and Guatemala (n=60) were provided LPG stoves, free fuel for 2 months and behaviour change messages. We tested the effectiveness of the behavioural messages by assessing the rate of exclusive LPG use monitored by temperature data loggers (Dots)[48 49] on stoves and acceptability through feedback from participants and field staff. Teams in Guatemala and Rwanda conducted FGDs with pilot household participants to revise behaviour change materials. In Peru, behavioural messages were developed and tested with non-pregnant adult women in the Cardiopulmonary outcomes and Household Air Pollution trial[13]; messages specific to pregnant women and new mothers were assessed through interviews and FGDs with pregnant women or mothers

| | Guatemala | India | Peru | Rwanda |
|---|---|---|---|---|
| **Table 1** | Formative research methods to design a behavioural intervention for the HAPIN trial | | | |
| Participant observations Cooking demonstrations | Participant observations of cooking activities in 36 homes with LPG and wood stoves, 2–3 hours in each home | N/A* | N/A* | Eighteen 2-hour LPG cooking demonstrations and blind food tasting with non-LPG users (participants did not keep LPG stoves or cylinders) |
| In-depth interviews | Eighteen interviews with women (primary cooks; 26–68 years of age) and six group interviews with three or four male participants | Twenty-five interviews, 11 in Nagapattinam and 14 in Kallakurichi (previously Villupuram; 23 female cooks, 2 men; six solid fuel users, 4 LPG users, 15 mixed fuel users; 21–65 years of age) | Seven interviews (six pregnant women, 1 new mother) | Fifty-four interviews with female primary cooks (14 LPG users, 22 non-LPG users, and repeat interviews with 18 of the same non-LPG users after an LPG cooking demonstration) |
| Key informant discussions | One informal interview with an LPG distributor in Jalapa and one informal interview with a stove manufacturer | Informal discussions with LPG distributors and managers, and local community members | Informal discussions with field staff native to Puno | Twelve informal interviews with local field staff who installed the LPG stove and delivered behavioural training during the pilot study |
| Focus group discussions | Nine FGDs of 5–6 participants (51 women; 2 men) | Two informal social group discussions (one in each site) with local villagers | One FGD, 7 participants (4 pregnant women, 3 new mothers) | Five FGDs to develop behaviour change materials (4 participants per group; 18–68 years of age), 4 FGDs to refine materials with pilot participants (2 FGDs after 1 month of LPG use, 2 FGDs after 2 months of LPG use; women 18–33 years of age; 0–2 children per household; 3–7 participants per group) |
| LPG stove pilot study | Behavioural messages reviewed on LPG stove installation and reinforced at LPG cylinder delivery visits in 60 households over a 3-month period | Behavioural messages delivered at LPG stove installation to the 20 pilot intervention households | N/A (messages and materials piloted through CHAP study)[13] | Behavioural messages and materials delivered to 40 pilot study households |

*Participant observations and cooking demonstrations were not conducted in Peru or India given widespread awareness of LPG and previous research in these areas.[32]

CHAP, Cardiopulmonary outcomes and Household Air Pollution; FGDs, focus group discussions; IDIs, in-depth interviews; LPG, liquefied petroleum gas.

with children under 2 selected according to the criteria explained above.

### Behavioural strategy development

After finalising the behavioural messages, a questionnaire and instruction sheet were developed using the COM–B model and BCW. These will guide the implementation of messages as part of a larger behaviour change strategy and will be used to monitor the effectiveness in achieving exclusive LPG use in the HAPIN trial.

### Ethics approval

The formative research protocol was reviewed and approved by the Institutional Review Boards of Emory University (00089799); the Bloomberg School of Public Health, Johns Hopkins University (00007464); Asociación Benéfica PRISMA in Peru (CE3571.16); Sri Ramachandra Institute of Higher Education and Research (IEC-N1/16/JUL/54/49); the Indian Council of Medical Research–Health Ministry Screening Committee (5/8/4–30/(Env)/Indo-US/2016-NCD-I); Universidad del Valle de Guatemala (146-08-2016); the Guatemalan Ministry of Health National Ethics Committee (11–2016); the London School of Hygiene and Tropical Medicine (11 664–2); and the Rwandan National Ethics Committee (No. 148/RNEC/2017). The HAPIN trial is registered with clinicaltrials.gov (NCT02944682).

### Patient and public involvement

The formative research reported in this manuscript was explicitly designed to engage community members at all four research sites in the design of an LPG intervention

and behavioural reinforcement package to be implemented in the main HAPIN trial. Community members were involved in the initial identification of messages for promoting exclusive LPG use, as well as the refinement of the materials and methods for delivering those messages.

## Data analysis

Qualitative data from IDIs and FGDs were analysed using thematic analysis, which is flexible and atheoretical and can be applied across a range of qualitative methodologies.[50] Thematic analysis assists in the identification and organisation of patterns in the data.[50] We used both an inductive and deductive approach. In Guatemala, data were transcribed and coded using HyperRESEARCH Software (Randolph, Massachusetts, USA). Other country sites used Microsoft Excel (2016) to track themes and relevant quotes. Each country site analysed their own data, which the first authors compiled for this manuscript.

## RESULTS

Table 1 summarises the formative research activities conducted in each research site.

## Formative research results

We identified nine main themes that influence exclusive LPG use: (1) perceived disadvantages of solid fuel stoves, (2) family influences on cooking decisions, (3) traditional cookware and stoves on which they are used, (4) traditional foods and preferences for stoves used to prepare them, (5) other non-food related reasons for cooking, (6) heating needs, (7) previous awareness and experience with LPG, (8) safety concerns and (9) cost and availability of LPG. We provide a brief description of the themes below; specific sub-themes are summarised in table 2.

## Reasons for abandonment of solid fuel stoves

Participants identified several disadvantages of solid fuel stoves, which suggest potential reasons for abandonment of traditional stoves.

### Family preferences for cooking practices

Many participants mentioned that family preferences influenced decisions about which stove to use for cooking tasks.

### Cookware

In Guatemala, Peru and Rwanda, participants raised concerns that LPG stoves would not accommodate the pots and cookware they needed to cook local staple foods.

### Traditional food

All sites, except India, identified traditional foods that people preferred to prepare with solid fuel stoves. Participants in Peru reported preferring to make a steamed quinoa bread (*quispiño*) with the traditional stove. In Guatemala and Rwanda, participants preferred to cook beans on the open fire because they believed beans

cooked more slowly on LPG stoves. Additionally, in Rwanda, *ugali*, or cassava bread, is difficult to make on an LPG stove because of the force required to stir the dough, which could cause the burner grate to break.

### Other uses of the stove

Traditional stoves are often used for purposes other than family meals, such as heating water for bathing during cold months. In Peru, people also commonly cook food for pigs and dogs. In Rwanda, open fires are sometimes used to make sorghum beer in large pots.

### Home heating needs

Warmth emanating from the traditional stove was valued during cold months in Guatemala and Peru, and to a lesser extent in India. In Guatemala, participants used the traditional stove for space heating but said they would forgo this if they had free LPG. In Peru, participants described using extra layers of clothing instead of lighting their traditional stove for heat.

### LPG awareness

In Peru and India, where governmental campaigns are actively promoting LPG nationwide, LPG awareness was much higher than that in Guatemala and Rwanda, where no national LPG campaigns currently exist. Owning an LPG stove in India was considered highly aspirational.

### LPG fears and safety

Participants reported some fears and concerns about LPG stove and cylinder safety, such as leaks, explosions, burns and child safety. Several participants in Peru and Guatemala reported a lack of trust in the safety and reliability of products from some LPG companies. In India, participants' concerns about the safety of LPG were described as minimal and acceptable in light of other LPG benefits.

### LPG cost, supply and distribution

LPG refill costs were major barriers in all sites. Distance and inaccessibility of households also limited LPG cylinder refills. While the LPG market in India is extensive and highly regulated, there are fewer governmental controls and LPG sale points in Guatemala, Rwanda and Peru.

## Developing behavioral messages for the HAPIN trial

We mapped formative research findings onto the COM–B and TDF domains and developed behavioural messages to address identified themes and domains (table 3). Factors related to capabilities and skills will be addressed using how-to materials and training, whereas factors related to motivation will be targeted with appeals to emotions such as trust, security and conscious decision-making. Factors related to opportunity and context address physical opportunities (providing prompt gas delivery and stove repair) as well as social opportunities (educating other members in the home to use the LPG stove) will be integrated into trial procedures.

**Table 2**  Summary of qualitative findings according to identified themes across study sites

| | Guatemala | India | Peru | Rwanda |
|---|:---:|:---:|:---:|:---:|
| **1. Perceived disadvantages of solid fuel stoves** | | | | |
| Smoke is physically irritating | X | X | X | X |
| Solid fuel stoves dirty kitchens, cookware, clothes and hands | X | X | X | X |
| Collecting and cooking with solid fuels requires time and energy costs | X | X | X | X |
| Monetary costs of solid fuel | X | | | X |
| Fear of snakes and environmental hazards when collecting fuel | X | X | X | X |
| Difficulty collecting and lighting wet solid fuel | | X | X | |
| **2. Family influences on cooking practices** | | | | |
| Family complaints that food gets cold quickly with LPG | X | | X | |
| Family complaints that food cooked with LPG lacks flavour | | | X | |
| Family preference for food cooked with LPG because food does not taste like smoke | | | | X |
| Family preference for LPG because food cooks faster | X | X | X | X |
| Family perception that LPG represents modernity | | X | | X |
| Husbands believe smoke harms their wives, but not husbands who do not cook | X | | | |
| **3. Cookware** | | | | |
| Belief that commonly used clay pots cannot be used on LPG stoves | X | | X | |
| Large, flat griddle required for tortillas | X | | | |
| Large pots required to cook staple foods | X | | | X |
| Meat, fish and vegetables commonly roasted on open fires | | | | X |
| **4. Traditional food** | | | | |
| Perception that some traditional dishes taste better when cooked with solid fuel | X | | X | X |
| Preference to cook food with solid fuel for family festivities and special occasions | | X | X | X |
| Preference to cook beans with solid fuel | X | | | X |
| **5. Other stove uses** | | | | |
| Heating water for bathing and washing | X | X | X | X |
| Cooking food for animals | | | X | |
| Making alcoholic beverages | | | | X |
| **6. Home heating needs** | | | | |
| Warmth from traditional stove viewed as beneficial during cold months | X | | X | |
| **7. LPG awareness** | | | | |
| Active governmental LPG campaigns have achieved high LPG awareness | | X | X | |
| Low LPG awareness in countries that lack governmental LPG campaigns | X | | | X |
| **8. LPG fears and safety** | | | | |
| Fear of LPG leaks and explosions or fires | X | | X | X |
| Fear of improperly attaching regulator and hose to the LPG cylinder | | | X | X |
| Fear of LPG-related burns | | | X | X |
| Concerns for child safety | X | | X | X |
| Mistrust of LPG providers | X | | X | |
| **9. LPG cost, supply and distribution** | | | | |
| LPG refills perceived as expensive | X | X | X | X |
| Large and highly regulated governmental LPG market | | X | | |
| Fewer governmental controls on LPG market | X | | X | X |
| Lack of LPG sale points and delivery capability in study areas | X | | X | X |
| Households are difficult to access (large distances between homes, lack of roads for transport) | X | | X | |

LPG, liquefied petroleum gas.

**Table 3** Themes, behavioural messages and strategies based on the BCW developed during formative research for the HAPIN trial

| Themes | Behavioural messages | COM–B/TDF domain* | Strategies and *related intervention functions* |
|---|---|---|---|
| 1. Perceived disadvantages of solid fuel | Using gas prevents discomfort (by reducing smoke)<br><br>Gas can be used in all seasons/weather<br><br>Using gas is easy<br><br>Gas eliminates smoke in the home<br><br>Gas keeps hands, clothes, pots and kitchens cleaner<br><br>With gas you do not need to collect or buy solid fuel<br><br>Gas will not make holes in thatch/aluminium roof | *Motivation*/reinforcement; emotions; optimism; beliefs about consequences<br><br>*Opportunity*/environmental context and resources | ▶ Emphasise disadvantages of traditional stoves to encourage abandonment of solid fuel<br>▶ *Education; Persuasion; Environmental restructuring* |
| 2. Family influences | Tips for addressing concerns of household members<br><br>Tips for addressing concerns of friends/neighbours<br><br>You can keep foods hot, or reheat quickly, after cooking them with gas<br><br>Using gas saves money and time | *Motivation*/emotion; beliefs about capabilities; optimism<br><br>*Opportunity*/social influences | ▶ Target behavioural interventions to all household members, not just primary cooks<br>▶ *Education; Persuasion; Enablement* |
| 3. Cookware | Using clay and other pots on the gas stove<br><br>How to cook large quantities of food with gas<br><br>How to roast on an LPG stove | *Capability*/knowledge; skills | ▶ Stove use demonstrations<br>▶ Guatemala and Rwanda: provide cookware to enable typical cooking behaviours<br>▶ *Training; Modelling* |
| 4. Traditional food | It is possible to cook beans on an LPG stove<br><br>How to cook traditional dishes with gas<br><br>How to enhance food flavour without solid fuel<br><br>How to make beer on an LPG stove<br><br>Practice makes perfect | *Capability*/knowledge; skills<br><br>*Motivation*/reinforcement; intentions; beliefs about capabilities; optimism | ▶ Guatemala and Rwanda: encourage soaking beans<br>▶ Rwanda: emphasise removing large pots from stove for forceful stirring<br>▶ *Education; Persuasion; Training* |
| 5. Other stove uses | Everything can be done with gas | *Motivation*/goals; reinforcement; intentions; beliefs about capabilities<br><br>*Opportunity*/environmental context and resources | ▶ Reassure households that LPG will be provided to meet all household cooking needs<br>▶ *Education; Persuasion; Incentivisation; Environmental restructuring* |
| 6. Home heating needs | How to stay warm when cooking with gas | *Motivation*/reinforcement; intentions; beliefs about consequences | ▶ Emphasise that no stove should be used for heating home<br>▶ Emphasise other LPG benefits as trade-offs for lack of heat<br>▶ *Education; Persuasion* |
| 7. LPG awareness | How to use LPG stove (turn off, turn on, open and close the gas)<br><br>How to regulate the flame, to prevent burning food and to save gas<br><br>How to clean stove | *Capability*/knowledge; skills; memory, attention and decision processes | ▶ Hands-on training on stove operation<br>▶ *Education; Training* |

Continued

**Table 3** Continued

| Themes | Behavioural messages | COM–B/TDF domain* | Strategies and *related intervention functions* |
|---|---|---|---|
| 8. LPG fears and safety | Gas is natural, like wood; the smell added to it is unpleasant to alert leaks, but not toxic | *Capability*/knowledge; skills; reinforcement; memory, attention and decision processes | ▶ Provide training on gas safety; provide phone numbers for project staff if leak detected or stove in need of repair; respond to household fears around gas use |
| | How to avoid burns | *Motivation*/emotion; beliefs about capabilities | ▶ *Education; Persuasion; Training; Environmental restructuring; Enablement* |
| | Child safety | | |
| | If used correctly, LPG stoves are completely safe | | |
| | How to check for and respond to a leak (soapy water) | | |
| | How to change the cylinder | | |
| | Explaining reasons why LPG brand can be trusted | | |
| | Millions of people use LPG stoves with no problems | | |
| | Who to call if there is a problem | | |
| | Where/how to get technical support | | |
| | How to store stove and gas cylinders properly | | |
| 9. LPG costs, supply, and distribution | Anticipating when gas will run out (cylinder check) | *Capability*/knowledge; skills; reinforcement; memory, attention and decision processes | ▶ Provide phone numbers for project staff if need gas refill; at installation instruct on secure storage of stove and cylinders |
| | What to do when you need a gas refill (including when and who to call) | *Opportunity*/environmental context and resources | ▶ *Enablement; Education; Persuasion; Training; Environmental restructuring* |
| | Security measures to prevent theft | | |

*Examples of theoretical domains are provided, but are not exhaustive. BCW, Behaviour Change Wheel; COM–B, Capability, Opportunity, Motivation–Behaviour; HAPIN, Household Air Pollution Intervention Network; LPG, liquefied petroleum gas; TDF, Theoretical Domains Framework.

Using the BCW, we identified seven intervention functions we will use to deliver messages that might lead to exclusive LPG use: *education* to increase knowledge and confidence in safe LPG use, *persuasion* to promote positive feelings about LPG benefits, *training* to enable LPG use to meet household needs, *environmental restructuring* to situate the LPG stove in kitchens that are free of smoke, modelling LPG stove use through hands-on training such as demonstrations of stove operation, *incentivisation* by providing free LPG gas and *enablement* by providing prompt LPG delivery and stove repairs. Because behavioural reinforcement visits are intended to be positive reinforcements and are not meant to be coercive or to induce negative emotions, two intervention functions, *restriction* and *coercion*, are not pertinent.

### Protocol for delivering behavioral strategies during the HAPIN trial

#### Stove package and equipment

Free, unlimited supply of LPG will be provided to intervention arm participants in the HAPIN trial to incentivise exclusive LPG use. To ensure constant supply (intervention function: *environmental restructuring*), two LPG cylinders will be provided. In Guatemala and Rwanda, the cylinders will have T-valve regulators with a flow switch that can be toggled to a second full tank when the first is empty. In India and Peru, families will be instructed to manually move the regulator between the cylinders. In Guatemala, the two cylinders will be installed outside the kitchen with a protective barrier. In Rwanda and Peru, cylinders will be installed inside, due to potential theft and freezing temperatures, respectively. Guatemala, Peru and Rwanda will provide a three-burner stove and India will provide a two-burner stove, deemed to fulfil cooking needs during formative research. In Rwanda and India, tables will be provided for the LPG stove; in Peru and Guatemala, table-height stoves will be provided. To assure that traditional foods will be cooked on the LPG stove, the Guatemalan stove will include a griddle (*comal*) for cooking tortillas and households will receive a set of enamel pots. In Peru, households will be instructed to grease clay pots before using on the gas stove to prevent cracking. Households in Rwanda will be given a roasting appliance for grilling meats and vegetables.

#### Stove use pledge

When the LPG equipment is installed in intervention households in the HAPIN trial, field staff will ask all

household members to be present and will administer a verbal pledge. By completing the pledge, participants will affirm that (1) they understand the study goals of reducing smoke exposures and achieving exclusive LPG use, (2) any type of food can be cooked with LPG, (3) the LPG stove should be used only for household cooking needs, (4) the stove/cylinder should not be sold or rented, (5) HAPIN staff are available to help with any challenges related to the LPG stove and (6) all household members intend to use the LPG stove exclusively (intervention function: *persuasion*).

### Stove installation and training

At the LPG stove installation visit in the HAPIN trial, trained field technicians will provide training on: (1) lighting/adjusting the gas flame, (2) cleaning/maintaining the stove, (3) detecting/responding to gas leaks, (4) requesting cylinder refills and stove/cylinder repairs, (5) safe handling/use of cylinders and regulators, (6) benefits of LPG and (7) disadvantages of solid fuel. In India, authorised technicians will collaborate with HAPIN staff to provide this training. In Rwanda, households will be required to pass a certification exam, demonstrating their ability to correctly perform the steps for safe stove and cylinder use before LPG stove installation (intervention functions: *education and training*).

### Printed materials (calendars, booklets, pamphlets, posters)

At stove installation, study staff in Guatemala, Peru and Rwanda will use a flipchart to deliver behavioural messages to participants. Participants in Guatemala, Rwanda and Peru will also receive a printed guide, calendar and/or poster containing pictorial and written representations of the behavioural messages to keep in their homes. In India, a flyer showing that a range of potential cooking tasks should be performed with LPG instead of the traditional stove will be left with households. Because LPG is highly aspirational and increasingly available through governmental programmes in India, printed materials on LPG benefits and traditional stove disadvantages will not be given to households to minimise unintended dissemination to control households.

### Videos

In Guatemala and Rwanda, videos on safe stove and cylinder use, how to check for and respond to a gas leak, cleaning the gas stove and cooking beans and other local dishes will be shown on a tablet to participants. Videos prepared in Rwanda will feature testimonials from both male and female LPG users, given formative research findings that men have a large influence over household decision-making.

### Monitoring and reinforcing LPG use during the HAPIN trial

The following sections outline how we will monitor behavioural strategy effectiveness to achieve exclusive LPG use and how we will identify households that continue to use solid fuel for behavioural reinforcement visits in the HAPIN trial.

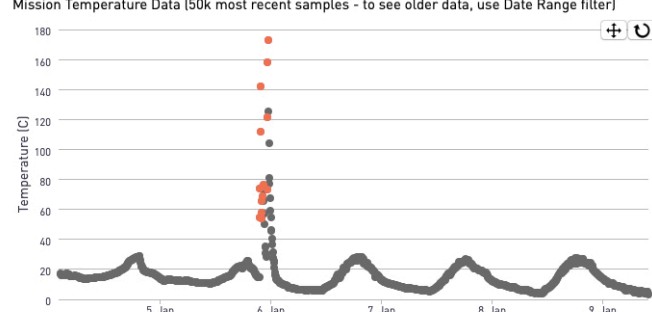

**Figure 1** Geocene Dot data from one household showing a flagged cooking event with a rapid temperature increase.

### Stove use monitoring

Temperature data loggers known as Dots (Geocene, Vallejo, CA, USA) will be installed on all solid fuel stoves in intervention households.[48] The Dots data loggers will be placed near or within the combustion zone to provide clear temperature signals at 5 min sampling intervals. Using a mobile application, field teams will download data from each Dot every 2 weeks to be analysed on a secure cloud-based server. A deterministic algorithm will be used to identify rapid, sustained temperature increases, which will be flagged as traditional stove use events (figure 1). Every week, local field staff, with periodic oversight by the BEC, will review households with flagged traditional stove use events based on the Dot data. They will use this data to schedule reinforcement visits, as described below.

### Observations of traditional stove use

Intervention households may build new makeshift fires that are not monitored by a temperature sensor or may remove the sensors from monitored stoves if they want to cook with solid fuel. Therefore, we will incorporate direct observations of traditional stove use into study activities. Field staff will conduct these observations at least once and up to three times per month in all intervention households. Using a checklist, staff will look for signs of recent traditional stove use, such as use during the visit, fresh ashes, hot embers, stoves that are warm to the touch, fresh blackening on walls, or lingering smoke.

### Data tracking

Monthly LPG use will be monitored by LPG delivery staff, based on the frequency of refills provided to households. In households using less LPG than average, staff will assess whether supplemental solid fuel stove use is occurring. In households with high LPG usage, staff will confirm that the LPG stove is being used properly, that is, not shared with neighbours, not used to prepare food for sale, appropriate flame settings to avoid fuel waste and lids on pots. In Guatemala, Rwanda and Peru, HAPIN staff will deliver LPG cylinder refills. In India, local LPG companies will deliver LPG with oversight from HAPIN staff.

## LPG use reinforcement visits

Field staff will review the Dot data, observations and LPG data described above to identify HAPIN trial households using solid fuel stoves. Within 1 week of identifying the household, field staff will visit to reinforce abandonment of the solid fuel stove and promote exclusive LPG use. A questionnaire will be administered to elicit concerns or challenges related to the LPG stove, allowing field staff to address their specific problems. For example, if participants mention that they cannot cook traditional dishes on the LPG stove, the field technician may show a how-to video or explain how to cook that dish using LPG. If participants are anxious about switching the valve between LPG cylinders, the staff will demonstrate the process and coach the participant to perform the procedure. In Guatemala, behavioural staff will observe cooking and conduct demonstrations with intervention participants who use their traditional stove. In Rwanda, LPG testimonial videos featuring local families will be used to demonstrate benefits of LPG use.

## Questionnaire on perceptions of LPG

A questionnaire on household perceptions of the LPG stove will be administered twice during pregnancy and twice after childbirth with HAPIN intervention households. The questionnaire, based on the COM–B and BCW will uncover additional barriers and facilitators related to LPG use that could be incorporated into behavioural messaging as the main trial progresses.

## DISCUSSION

We identified nine potential influencers of exclusive LPG use at the household level, including perceived disadvantages of solid fuel, family preferences, cookware, traditional foods, non-food-related cooking, heating needs, LPG awareness, safety and cost and availability of fuel. These factors are similar to those found by Puzzolo et al[20] in their systematic review of cleaner fuel use. Our study is unique because we used formative research grounded in behaviour change theory to design behavioural strategies to promote exclusive use of LPG in intervention households and solid fuel stove abandonment for the HAPIN trial.

Too often, interventions assume that introduction of cleaner fuels and technologies alone will be enough to eliminate HAP exposure. However, without a clear understanding and targeted approach to address cooking behaviours, family dynamics and environmental constraints, households often resume use of solid fuel stoves for some or most of their cooking needs.[14] Our research was guided by an overarching set of common principles generalisable across contexts, but also uncovered contextual differences requiring tailored behavioural approaches. All behavioural strategies are intended to increase LPG adoption among intervention households with some contextualisation to local conditions (eg, climate differences, cooking practices) during the HAPIN trial.

Achieving exclusive use of new cooking technologies requires that study participants abandon, or deimplement, the old cooking technology. Such deimplementation has been used in behavioural intervention studies to change low-value practices or harmful behaviours.[44] Everett Rogers' diffusion of innovation theory suggests that households are more likely to abandon an old technology in favour of a new one when the new device has relative advantages over the old one, is compatible with local practices and is not too complex to use.[51] During our formative phase, we provided 40–60 homes in Guatemala, India and Rwanda with LPG stoves for 2 months to test acceptability, appropriateness and feasibility of LPG stove and fuel use.[47] This initial phase enabled us to identify local perceptions of the relative advantages of LPG over traditional stoves, local practices that needed to be framed as compatible with LPG and how to reduce complexity of the LPG technology that we incorporated into training and behaviour change strategies in each research site.[51]

Our formative research highlighted several areas that build on efforts of previous cookstove trials. For example, self-reported stove use has been shown to overestimate use of the improved or cleaner stove and underestimate continued use of the solid fuel stove.[52] Other studies have used temperature data loggers to monitor the cleaner stove but did not monitor all solid fuel stoves in the home, which limits the ability to estimate stove stacking.[53] To better understand stove use and stacking behaviours, our study applies temperature data loggers on all traditional stoves with observations of traditional stove use at monthly home visits. Real-time data summaries will allow continuous follow-up during the trial, flagging households using traditional stoves. Field staff will visit homes to troubleshoot potential LPG stove problems or other barriers and reinforce exclusive LPG use. Observations and responses to questionnaires on LPG perceptions and use will inform continuous adaptation to behavioural messages to maximise LPG adoption.

We designed our behavioural messaging to emphasise immediately visible disadvantages of cooking with solid fuels such as dirty kitchens and physical discomfort to encourage abandonment, based on our formative research that suggested these disadvantages were more tangible than long-term health effects. Other studies have also found focusing on health risks to be less effective.[32] Addressing context-specific fears and concerns, grounded in theory, may prove to be more effective than solely addressing capabilities, or skills training, on how to use the LPG stove. While skills training is essential for adoption of unfamiliar technologies, additional behaviour change messages that target motivations and opportunities among all household members may encourage a more complete household transition to exclusive LPG use. The TDF describes motivations, or social norms, as an essential part of designing behavioural interventions, and household members may either support or thwart the use of a new stove technology. Because the

trial will provide free LPG, we will target opportunities by addressing environmental resources and context. This will assure that participants will be able to use the LPG stove for all purposes, including cooking animal fodder and brewing beer, which is uncommon when people pay for their own fuel.[20]

Cost remains one of the main drivers of cleaner fuel adoption.[20 22 23] Both monetary and time costs of obtaining cleaner fuel are frequent barriers to adoption.[23 46] In many rural areas, LPG cylinders are not delivered to homes, requiring families to travel long distances to procure fuel.[54] The HAPIN trial will provide 18 months of free fuel delivered to intervention households to overcome economic and transportation barriers and promote exclusive LPG use. Our formative research highlighted additional factors unrelated to cost that we hypothesise must also be addressed to achieve exclusive LPG use, such as reinforcing perceived disadvantages of cooking with solid fuel, addressing fears of LPG, fulfilling non-cooking needs for stove use such as heating and preparing animal fodder, and ensuring that LPG cooking is compatible with traditional foods. An additional influencer of clean fuel adoption and sustained use is the powerful role of market forces that generate adequate supply and demand activities to meet the needs of households that wish to use cleaner fuels. Because the HAPIN trial will provide free fuel, we did not explore market forces during the formative research, but an aim of our future work is to understand supply and demand for LPG in the HAPIN trial sites with the goal of facilitating post-trial access to clean fuels.

Several potential limitations should be noted. First, we may have missed important contextual factors during our formative research. For example, in multifamily households, one LPG stove per household may not be sufficient to meet everyone's needs. Additionally, positive behavioural reinforcements may not be sufficient for intervention households that refuse to abandon solid fuel stoves. The complexity of changing cooking behaviours is one of the greatest challenges in stove adoption studies.[29 55 56] Second, our monitoring strategies may not accurately flag traditional stove use, which may result in unnecessary behavioural reinforcement visits to compliant households. Third, while we will track monthly LPG usage to assure that LPG households are requesting refills, LPG usage varies based on differences in household cooking tasks, family size and other factors. Thus, we may incorrectly flag low LPG users for reinforcement. However, our extensive monitoring of stove use through observations, stove use questionnaires and Dot data loggers will allow triangulation and offer insights into reasons for use and non-use of the LPG intervention over the 18-month trial. Lastly, our formative research, behaviour change intervention and monitoring plans are extensive and may not be feasible in all contexts. The HAPIN trial is not designed to determine which aspects of the intervention are critical for achieving exclusive LPG use, but rather to do everything possible to achieve exclusive use. Future research will be needed to test which components, that is, cost removal, home delivery, stove use training, behavioural reinforcement, and so on, are necessary and sufficient to achieve exclusive LPG use.

## CONCLUSION

Achieving the highest possible exclusive LPG use among intervention households is essential for understanding the potential exposure reductions and health benefits that an LPG cooking intervention can provide. While our approach is more intensive than a real-world LPG promotion programme, our formative research results provide valuable insights on how to develop, implement, monitor and evaluate theory-informed behavioural strategies to promote LPG adoption and exclusive use. Strategies for promoting and monitoring exclusive LPG use are important not only to understand the impact of LPG adoption within trials, but also to sustain use in broader programmes and promotional campaigns. While the behavioural components of the intervention were designed in the context of the HAPIN trial, the methods and lessons learnt may provide insights for achieving sustained, exclusive use of cleaner fuels when delivered programmatically at scale.

**Author affiliations**
[1]Division of Pulmonary and Critical Care Medicine, Johns Hopkins School of Medicine, Baltimore, Maryland, USA
[2]Center for Global Non-Communicable Disease Research and Training, Johns Hopkins School of Medicine, Baltimore, Maryland, USA
[3]Nell Hodgson Woodruff School of Nursing, Emory University, Atlanta, Georgia, USA
[4]Department of Disease Control, Faculty of Infectious and Tropical Diseases, London School of Hygiene and Tropical Medicine, London, UK
[5]Centro de Estudios en Salud, Universidad del Valle de Guatemala, Guatemala City, Guatemala
[6]Fogarty International Center, National Institutes of Health, Bethesda, Maryland, USA
[7]Department of Environmental Health Engineering, Sri Ramachandra Institute for Higher Education and Research, Chennai, India
[8]Department of Public Health and Policy, University of Liverpool, Liverpool, UK
[9]Department of Environmental & Radiological Health Sciences, Colorado School of Public Health, Aurora, Colorado, USA
[10]Rollins School of Public Health, Emory University, Atlanta, Georgia, USA
[11]CRONICAS Center of Excellence in Chronic Diseases, Universidad Peruana Cayetano Heredia, Lima, Peru
[12]Department of International Health, Social and Behavioral Interventions, Johns Hopkins University Bloomberg School of Public Health, Baltimore, Maryland, USA

**Acknowledgements** The authors acknowledge that a multidisciplinary, independent Data and Safety Monitoring Board (DSMB) appointed by the National Heart, Lung, and Blood Institute (NHLBI) monitors the quality of the data and protects the safety of patients enrolled in the HAPIN trial. NHLBI DSMB: Nancy R Cook, Stephen Hecht, Catherine Karr, Joseph Millum, Nalini Sathiakumar (Chair), Paul K Whelton, Gail G Weinmann (Executive Secretary). Program Coordination: Gail Rodgers, Bill & Melinda Gates Foundation; Claudia L Thompson, National Institute of Environmental Health Science; Mark J Parascandola, National Cancer Institute; Danuta M Krotoski and Marion Koso-Thomas, Eunice Kennedy Shriver National Institute of Child Health and Human Development; Joshua P Rosenthal, Fogarty International Center; Conception R Nierras, NIH Office of Strategic Coordination Common Fund; Katie Kavounis, Dong-Yun Kim, Antonello Punturieri and Barry S Schmetter, NHLBI.

**Collaborators** Household Air Pollution Intervention Network (HAPIN) trial Investigators: Vigneswari Aravindalochanan, Dana Boyd Barr, Vanessa Burrowes, Alejandra Bussalleu, Devan Campbell, Eduardo Canuz, Adly Castañaza, Howard Chang, Yunyun Chen, Marilú Chiang, Maggie L Clark, Rachel Craik, Mary Crocker, Victor Davila-Roman, Lisa de las Fuentes, Oscar De León, Ephrem Dusabimana, Lisa Elon, Juan Gabriel Espinoza, Irma Sayury Pineda Fuentes, Sarada Garg, Dina

Goodman, Savannah Gupton, Meghan Hardison, Stella Hartinger, Phabiola M Herrera, Shakir Hossen, Penelope Howards, Lindsay Jaacks, Shirin Jabbarzadeh, Michael A Johnson, Abigail Jones, Katherine Kearns, Miles Kirby, Jacob Kremer, Margaret A Laws, Pattie Lenzen, Jiawen Liao, Amy E Lovvorn, Fiona Majorin, Eric McCollum, John McCracken, Julia N McPeek, Rachel Meyers, Erick Mollinedo, Lawrence Moulton, Krishnendu Mukhopadhyay, Luke Naeher, Abidan Nambajimana, Durairaj Natesan, Florien Ndagijimana, Azhar Nizam, Jean de Dieu Ntivuguruzwa, Aris Papageorghiou, Ricardo Piedrahita, Ajay Pillarisetti, Naveen Puttaswamy, Sarah Rajkumar, Usha Ramakrishnan, Rengaraj Ramasami, Davis Reardon, Barry Ryan, Sudhakar Saidam, Sankar Sambandam, Jeremy A Sarnat, Suzanne Simkovich, Sheela S Sinharoy, Kirk R Smith, Kyle Steenland, Damien Swearing, Ashley Toenjes, Lindsay Underhill, Jean Damascene Uwizeyimana, Viviane Valdes, Kayla Valentine, Amit Verma, Lance Waller, Megan Warnock, Wenlu Ye, Bonnie Young.

**Contributors** KW and LT led the writing of the manuscript. ZS and GR managed data collection and analysis in Rwanda. MH, ADA and LT managed data collection and analysis in Guatemala. GT and KB managed data collection and analysis in India. KW, SAH and WC managed data collection and analysis in Peru. AQ, EP, JPR, TFC, JJM and JPR provided overall guidance to study implementation. All authors contributed to the cross-site synthesis of findings, development of the study protocol and writing and revision of the manuscript.

**Funding** The HAPIN trial is funded by the US National Institutes of Health (cooperative agreement 1UM1HL134590) in collaboration with the Bill & Melinda Gates Foundation (OPP1131279). The findings and conclusions in this report are those of the authors and do not necessarily represent the official position of the US National Institutes of Health, Department of Health and Human Services or the Bill & Melinda Gates Foundation.

**Competing interests** None declared.

**Patient and public involvement** Patients and/or the public were involved in the design, or conduct, or reporting or dissemination plans of this research. Refer to the Methods section for further details.

**Patient consent for publication** Not required.

**Provenance and peer review** Not commissioned; externally peer reviewed.

**Data availability statement** Deidentified participant data are available upon reasonable request. Contact Kendra Williams (kendra.williams@jhu.edu) for Peru data, Lisa Thompson (lisa.thompson@emory.edu) for Guatemala data, Gurusamy Thangavel (thangavel@ehe.org.in) for India data or Ghislaine Rosa (ghislaine.rosa@lshtm.ac.uk) for Rwanda data. For data reuse conditions, please contact Lance Waller (lwaller@emory.edu).

**ORCID iDs**
Kendra N. Williams http://orcid.org/0000-0001-9697-048X
J Jaime Miranda http://orcid.org/0000-0002-4738-5468

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
