## [Reviewer comments · BMJ Open]

ARTICLE DETAILS

TITLE (PROVISIONAL)	Designing a comprehensive behavior change intervention to promote and monitor exclusive use of liquefied petroleum gas stoves for the Household Air Pollution Intervention Network (HAPIN) trial
AUTHORS	Williams, Kendra; Thompson, Lisa; Sakas, Zoe; Hengstermann, Mayari; Quinn, Ashlinn; Díaz Artiga, Anaité; Thangavel, Gurusamy; Puzzolo, Elisa; Rosa, Ghislaine; Balakrishnan, Kalpana; Peel, Jennifer; Checkley, William; Clasen, Thomas; Miranda, J. Jaime; Rosenthal, Joshua; Harvey, Steven

VERSION 1 – REVIEW

REVIEWER	William D Evans The George Washington University, USA
REVIEW RETURNED	11-Mar-2020

GENERAL COMMENTS	This is an interesting and valuable study on formative research for the HAPIN trial. The manuscript requires a few minor revisions prior to publication:  1. Page 6, Line 45, use a word like 'abandon' rather than 'dis-adoption' which is awkward. 2. Bottom page 6/top page 7, discussion focuses on market-based factors (supply/demand) but doesn't discuss the role of markets. The authors should consider and at least comment (perhaps in the discussion) the powerful role of market forces in clean stove adoption. Without adequate supply and demand generation activities, adoption of LPG and similar clean technologies is very unlikely. 3. No research questions or hypotheses are stated. Did the study have any, and if so please add them near end of the introduction. Or explain if not. 4. All of the BEC strategies (pp. 8-9) seem to relate to use, but this doesn't address market forces (comment above). Use is unlikely without addressing supply/demand factors. 5. The design of the study is not clear (pp. 11-12). Was this based on a convenience sample? Any form of randomization or control for selection bias? 6. There is no description of the data analysis. This needs to be added in some detail, including use of any software. 7. In the discussion, I think there needs to be acknowledgement that the research does not address market forces, either in the data collected or study design. This is an important limitation. The research has substantial value, but without addressing supply and demand issues, it is very hard to envision widespread adoption of LPG, which has costs and benefits that need to be addressed through effective market management.
--

REVIEWER	Rob Bailis Stockholm Environment Institute - US Center USA
REVIEW RETURNED	15-Apr-2020

GENERAL COMMENTS	This paper discusses formative research carried out in the early stages of the HAPIN trial. The objective as I understand it is to identify factors that influence fuel choice in study communities across four countries and to identify behavioral change strategies that will support the exclusive use of LPG. The article is well-organized and clearly written. The authors provide a clear theoretical framework and present a thorough overview of the broader study that this preliminary work supports. The article deserves publication after a few minor revisions, which I describe below (page numbers refer to the numbers on the upper left of the pdf file I reviewed). p. 9 lines 28/29: The authors note that "few programs and research studies have integrated behavioral components into their campaigns, instead focusing on short-term adoption of the new technology". I don't think this is a fair assessment. The authors could acknowledge that the GACC/Cooking Alliance has been working on behavior change for several years (e.g. https://www.cleancookingalliance.org/market-development/demand-creation/behavior-change-communication.html). USAID's TRACTION program funded several BC-focused projects back in 2011-2014. So, it's not as if BC hasn't been integrated into clean cooking efforts. It may be more accurate to say that they haven't seen much success. p. 11 lines 45-50: I found the discussion of TDF "domains" and "constructs" lacking in detail. How do domains relate to constructs? The authors note the constructs are "synthesized from multiple theoretical models" (but cite no sources at that point) and then move on to discuss the BCW approach. Are the constructs important? If yes, why raise them here, but not mention them again? If they're not important, then don't bring them up at all... p. 12 line 17: It would be helpful to have a little more information about the study sites. Are these all rural communities? Very remote, or accessible? Off-grid? Etc... p. 12 line 29: Why combine primary use and exclusive use here? This entire study is based on achieving exclusive use and differentiating the extent of exclusive use at baseline seems important. p. 12 line 38-40: At this point it would be helpful for readers if the authors refer readers to Table 1 so they are aware of the number of FGDs and IDIs they conducted. Also, the authors should explain how FGD and IDI participants were selected. p. 13 line 38: The authors note that FGDs in India sought info that would minimize resentment and contamination bias in control HHs". Why was this done in India - is there no risk of resentment in other places?
---

	p. 24 line 52: The authors note that participants pledge several things including that they "understand the study goals of reducing smoke exposures". However, "Guiding Principle 5" said that the researchers would "avoid emphasizing potential health benefits of LPG to minimize the risk of introducing bias". Does the participants' pledge contradict this guiding principle?
--	--

VERSION 1 – AUTHOR RESPONSE

Reviewer: 1

Reviewer Name: William D Evans

Institution and Country: The George Washington University, USA

Please state any competing interests or state 'None declared': None declared

Please leave your comments for the authors below

This is an interesting and valuable study on formative research for the HAPIN trial. The manuscript requires a few minor revisions prior to publication:

1. Page 6, Line 45, use a word like 'abandon' rather than 'dis-adoption' which is awkward.

- We made changes throughout the text to replace dis-adoption with abandon/abandonment. We changed dis-adoption to de-implementation on page 29, since the de-implementation of “unhealthy” practices is an important area in implementation research.

2. Bottom page 6/top page 7, discussion focuses on market-based factors (supply/demand) but doesn't discuss the role of markets. The authors should consider and at least comment (perhaps in the discussion) the powerful role of market forces in clean stove adoption. Without adequate supply and demand generation activities, adoption of LPG and similar clean technologies is very unlikely.

- Thank you for this comment. We added a sentence at the top of page 7 to address this concern: “Markets that assure adequate supply to meet household demand are a critical need.” This formative research was designed specifically to inform behavioral strategies for the HAPIN trial, which is an efficacy trial that will provide free fuel and doesn't depend on market forces. However, we are working with local distributors to ensure LPG will be available for HAPIN households when they complete the study (although at this point households would have to purchase the LPG, which would likely influence adoption as the reviewer notes). We added an explanation of this at the bottom of page 31.

3. No research questions or hypotheses are stated. Did the study have any, and if so please add them near end of the introduction. Or explain if not.

- In this formative research to prepare for the HAPIN trial we explored strategies to promote exclusive use. We explain on page 8 that the objective of our paper was to conduct theory-guided formative research to develop locally-adapted behavioral strategies for promoting LPG use. The main HAPIN trial is hypothesis-driven, but the formative research did not have or test hypotheses.

4. All of the BEC strategies (pp. 8-9) seem to relate to use, but this doesn't address market forces (comment above). Use is unlikely without addressing supply/demand factors.

- We agree. Given that the HAPIN trial will provide free fuel, we did not address supply/demand factors in this formative research for the HAPIN trial. Our future work will aim to understand market forces and supply/demand factors in the HAPIN trial sites, with the goal of facilitating continued access to clean fuel for households after the trial ends.

5. The design of the study is not clear (pp. 11-12). Was this based on a convenience sample? Any form of randomization or control for selection bias?

- We added an explanation of the eligibility criteria for the pilot study to page 12. Since it was a pilot for the main trial, the primary goals were to test and revise the standard operating procedures for the main trial, estimate anticipated PM_{2.5} and black carbon levels, and test the effectiveness and acceptability of the behavioral strategies. Environmental analyses will be published separately. We have added an explanation of this on page 12.

6. There is no description of the data analysis. This needs to be added in some detail, including use of any software.

- We have added a data analysis section on page 14.

7. In the discussion, I think there needs to be acknowledgement that the research does not address market forces, either in the data collected or study design. This is an important limitation. The research has substantial value, but without addressing supply and demand issues, it is very hard to envision widespread adoption of LPG, which has costs and benefits that need to be addressed through effective market management.

- We agree. This manuscript serves to describe the behavior change strategies for the HAPIN trial, in which fuel will be provided and delivered for free. Definitely, market forces (availability, affordability, accessibility of LPG) are vital if households are to continue using LPG after the trial ends.

Reviewer: 2

Reviewer Name: Rob Bailis

Institution and Country:

Stockholm Environment Institute - US Center

USA

Please state any competing interests or state 'None declared': None declared

Please leave your comments for the authors below

This paper discusses formative research carried out in the early stages of the HAPIN trial. The objective as I understand it is to identify factors that influence fuel choice in study communities across four countries and to identify behavioral change strategies that will support the exclusive use of LPG. The article is well-organized and clearly written. The authors provide a clear theoretical framework and present a thorough overview of the broader study that this preliminary work supports. The article deserves publication after a few minor revisions, which I describe below (page numbers refer to the numbers on the upper left of the pdf file I reviewed).

p. 9 lines 28/29: The authors note that "few programs and research studies have integrated behavioral components into their campaigns, instead focusing on short-term adoption of the new technology". I don't think this is a fair assessment. The authors could acknowledge that the GACC/Cooking Alliance has been working on behavior change for several years (e.g. <https://www.cleancookingalliance.org/market-development/demand-creation/behavior-change-communication.html>). USAID's TRACTION program funded several BC-focused projects back in 2011-2014. So, it's not as if BC hasn't been integrated into clean cooking efforts. It may be more accurate to say that they haven't seen much success.

- This is an excellent point. We have modified that section (page 7) to focus on the lack of theoretical grounding in programs and research that have integrated behavioral components. We have also added a citation to the Clean Cooking Alliance Behavior Change Communication website.

p. 11 lines 45-50: I found the discussion of TDF "domains" and "constructs" lacking in detail. How do domains relate to constructs? The authors note the constructs are "synthesized from multiple theoretical models" (but cite no sources at that point) and then move on to discuss the BCW approach. Are the constructs important? If yes, why raise them here, but not mention them again? If they're not important, then don't bring them up at all...

- Very true. We removed the 84 constructs since we do not integrate them into our results/discussion and tried to clarify the text, providing additional references for the TDF validation (page 9).

p. 12 line 17: It would be helpful to have a little more information about the study sites. Are these all rural communities? Very remote, or accessible? Off-grid? Etc...

- We clarified that these are rural communities, gave the distance from main cities and provided population densities which include both rural and urban areas of the district (page 10).

p. 12 line 29: Why combine primary use and exclusive use here? This entire study is based on achieving exclusive use and differentiating the extent of exclusive use at baseline seems important.

- In India, we only had access to data on primary (not exclusive) LPG use. We have revised this to indicate that we refer to exclusive LPG use in Peru, Guatemala, and Rwanda and to primary LPG use in India (page 10).

p. 12 line 38-40: At this point it would be helpful for readers if the authors refer readers to Table 1 so they are aware of the number of FGDs and IDIs they conducted.

- We added a reference to Table 1 when explaining the IDIs (page 10) and the FGDs (page 11).

Also, the authors should explain how FGD and IDI participants were selected.

- We have added a description of how IDI and FGD participants were selected: "Participants were selected based on the following criteria: living in a rural community in the country-specific study site, female, between the ages of 18-68, and able to understand and provide consent. In each site, we aimed to include participants with and without previous knowledge and/or use of LPG. Teams in India and Guatemala also sought to include some men meeting the same criteria." (page 10-11).

p. 13 line 38: The authors note that FGDs in India sought info that would minimize resentment and contamination bias in control HHs". Why was this done in India - is there no risk of resentment in other places?

- Given large-scale governmental efforts for LPG promotion in India, desire for LPG was very high in our India site. This raised the concern that control households in the HAPIN trial may have a greater ability to adopt LPG if they were to see materials in intervention households about the benefits of LPG. Thus, in India we specifically designed FGDs to understand the minimum amount of material that we could provide to intervention households to achieve exclusive LPG use while preventing nearby controls from learning too much about LPG benefits and adopting LPG on their own through the generous governmental programs. This was less of a concern in Rwanda and Guatemala where LPG was largely unavailable, and in Peru where governmental subsidy programs were not sufficient to enable exclusive LPG adoption¹. We removed resentment because our control compensation packages (published separately²) were designed to minimize resentment. We have added a description to the paper that the extensive governmental support and ubiquity of LPG in India was the driving motivator to identify the minimum set of behavioral materials in India (page 12).

1. Pollard SL, Williams KN, O'Brien CJ, et al. An evaluation of the Fondo de Inclusion Social Energetico program to promote access to liquefied petroleum gas in Peru. *Energy Sustain Dev.* 2018;46:82-93.

2. Quinn AK, Williams K, Thompson LM, et al. Compensating control participants when the intervention is of significant value: Experience in Guatemala, India, Peru and Rwanda. *BMJ Global Health.* 2019;4(4).

p. 24 line 52: The authors note that participants pledge several things including that they "understand the study goals of reducing smoke exposures". However, "Guiding Principle 5" said that the researchers would "avoid emphasizing potential health benefits of LPG to minimize the risk of introducing bias". Does the participants' pledge contradict this guiding principle?

- We do not think that the pledge contradicts the guiding principle because we do not discuss the health benefits of reducing smoke exposure in the pledge. We reinforce that the study

goal is to reduce smoke exposure, which is a tangible benefit to the participants, but we do not tie that to health outcomes.

VERSION 2 – REVIEW

REVIEWER	William D Evans The George Washington University
REVIEW RETURNED	04-Jun-2020

GENERAL COMMENTS	The revised paper is improved and will make a valuable contribution to the literature. It should be accepted for publication after a careful copy edit is performed.
--

REVIEWER	Rob Bailis Stockholm Environment Institute - US Center
REVIEW RETURNED	19-Jun-2020

GENERAL COMMENTS	The authors did a nice job addressing both my comments and comments from the other reviewer. The revised paper is markedly improved and I have no further suggestions. I recommend it be published as is.
---